# Comparison of Direct and Indirect Detection of *Toxoplasma gondii* in Ovine Using Real-Time PCR, Serological and Histological Techniques

**DOI:** 10.3390/ani14101432

**Published:** 2024-05-10

**Authors:** Roberto Condoleo, Davide Santori, Erminia Sezzi, Salvatore Serra, Sara Tonon, Claudia Eleni, Antonio Bosco, Lucy Nicole Papa Caminiti, Maria Francesca Iulietto

**Affiliations:** 1Istituto Zooprofilattico Sperimentale del Lazio e della Toscana, 00178 Rome, Italy; davide.santori@izslt.it (D.S.); erminia.sezzi@izslt.it (E.S.); sara.tonon@izslt.it (S.T.); claudia.eleni@izslt.it (C.E.); lucy.papa-esterno@izslt.it (L.N.P.C.); mariafrancesca.iulietto@izslt.it (M.F.I.); 2Viterbo Local Health Unit, Veterinary Services, 01100 Viterbo, Italy; salvatore.serra@asl.vt.it; 3Department of Veterinary Medicine, Federico II University, 80137 Naples, Italy; boscoant@tiscali.it

**Keywords:** *Toxoplasma gondii*, sheep, PCR, serology, survey, lamb

## Abstract

**Simple Summary:**

Ingesting meat from lamb or mutton containing cysts of *Toxoplasma gondii* can lead to human infection, defined as toxoplasmosis. Many studies worldwide reported high antibody levels to *T. gondii* in sheep, but this does not always indicate that tissues destined for consumption contain infectious cysts. This study aimed to understand *T. gondii* occurrence in sheep muscles and the correlation between seropositivity and parasite presence. Samples from 349 sheep (blood, heart and diaphragm) underwent ELISA, real-time PCR and histological testing. Despite high seroprevalence rates, *T. gondii* DNA was found in only 3.7% of tested sheep (13/349), all adults. Histological tests did not detect *T. gondii* tissue cysts in any samples tested. The results agree with previously published research and might be influenced by the uneven distribution of tissue cysts in carcasses or muscle and by the analytical methods used.

**Abstract:**

*Toxoplasma gondii* is a zoonotic pathogen and the ingestion of tissue cysts by consumption of lamb or mutton has been identified as a possible cause of infection in humans. Many serological surveys in sheep have been performed, showing relevant serological rates; however, while the detection of antibodies indicates an exposure to *T. gondii*, this does not necessarily imply the presence of tissue cysts in edible tissue. The current study aims to provide further understanding on the occurrence of *T. gondii* in sheep muscles and the strength of correlation between serological positivity and presence of the parasite in sheep. From 349 sheep, samples (i.e., blood, heart and diaphragm) were collected and subjected to ELISA tests, real-time PCR and histological tests. Despite the high seroprevalence, *T. gondii* DNA was detected in the heart and/or the diaphragm from 13 out of the 349 tested sheep (3.7%); all were adults (13/191). Furthermore, the histological tests did not reveal the presence of *T. gondii* tissue cysts in any of the examined portions of interventricular septum. It should be considered that the likelihood of detecting genetic material of the parasite is probably influenced by the uneven distribution of the tissue cysts in the carcass as well as the methodology applied. The findings of this study support the importance of describing the uncertainty associated with the data used for risk assessment to reduce inaccurate estimation or risk overestimation.

## 1. Introduction

Toxoplasmosis is a zoonotic disease caused by a protozoan called *Toxoplasma gondii* (*T. gondii*), a ubiquitous parasite that occurs in most areas of the world [1]. Felids are the definitive hosts of *T. gondii* and can excrete oocysts containing the parasite infectious form (sporozoites) in the environment. Many species can be infected after the accidental ingestion of vital oocysts and act as intermediate hosts in which *T. gondii* can create tissue cysts that contain another infectious form (bradyzoites). Tissue cysts can be potentially found in every organ of the animal but tend to be more frequent in those mainly composed of nervous and muscle tissues (i.e., brain, heart, diaphragm) [2]. Bradyzoites can remain vital for long periods and other animals can become intermediate hosts by feeding on contaminated food (i.e., raw meat) derived from such infected animals; the cycle is complete when a felid ingests bradyzoites through contaminated food or oocysts present in the environment [3,4,5].

Most species are usually resistant to *T. gondii,* but some mammals, like humans and sheep, are particularly sensitive and may develop severe symptoms (e.g., neural disorders, abortion, fetal malformation, stillbirth) [6,7]. In addition, sheep are commonly raised through extensive systems, frequently grazing outdoors and, as a result, may have an increased risk of infection due to contamination of the environment with sporulated oocysts [8].

Among the pathogens associated with sheep farming, both in dairy and meat production, *T. gondii* represents a zoonotic pathogen of concern due to the severe associated public health implications. Ingestion of tissue cysts by consumption of lamb or mutton has been identified as a possible cause of infection in humans [4,9] and consumption of raw or undercooked sheep and goat meat is an important risk factor [1]. This exposure route should be taken into account given the production of over 10 million tons of meat from sheep worldwide in 2022. In Italy, during the last ten years, the number of sheep slaughtered ranged from 3,032,065 to 2,780,150 (2013–2022), with a steady production of over 29 tons of sheep meat in the last 3 years (2022–2024) [10].

The foodborne route is the main route of transmission for humans, and consumption of meat from pigs and sheep and vegetables was associated with a higher likelihood of infection [11,12]. As for the WHO structured expert elicitation, red meats (i.e., beef, small ruminants’ meat and pork) are estimated to cause 50% to 64% of foodborne toxoplasmosis cases at a global level; furthermore, small ruminants’ meat is estimated to cause over 40% of foodborne toxoplasmosis in the Eastern Mediterranean Region (EMR) subregions [13]. Sheep and goat meat are also considered important sources of infection in Southern European countries [8]. 

There are several methods available to detect *T. gondii* tachyzoites, bradyzoites and oocysts in or on food products but, at present, there are no specific regulations or ISO standards for the detection of *T. gondii* in any food matrix [14]; however, it is considered necessary to standardize detection methods [12].

Concerning live animals, diagnosis of *T. gondii* infection in livestock is performed by using direct and indirect methods [15]. Direct methods involve demonstrating the viability and infectivity of the parasite through the isolation of the organism via animal inoculation (using a bioassay method), inoculation of various biosamples in tissue cultures or detecting the parasite by amplifying specific nucleic acid sequences of the parasite through molecular methods, such as PCR. On the other hand, indirect methods include several serological tests that primarily indicate the presence of Toxoplasma-specific antibodies [16] in serum samples or in meat juice [17]. Presently, prevalence studies both in farmed and wild animals rely on serological analysis, with the ELISA test being the most cost-effective for large-scale investigations.

Many serological surveys in sheep over recent decades have been performed worldwide [4,18,19], showing relevant serological rates mainly due to extensive breeding. Although the detection of antibodies indicates an exposure to the parasite, this does not necessarily imply the presence of cysts in edible tissues. An extensive literature review conducted by EFSA described a moderate/low correlation between the serological status and the presence of *T. gondii* DNA in tissues [20]. However, their estimate was based on a few studies and a limited number of them are relevant considering the numerous factors that should be considered (e.g., sample size, age of the animal).

On the other hand, quantitative predictive models performed so far tend to be conservative and assume that muscle from all seropositive subjects is homogenously contaminated with a certain concentration bradyzoites [9,21,22]. Nevertheless, this simplification might imply a relevant level of uncertainty associated with the risk of toxoplasmosis caused by meat consumption. 

Farm management (i.e., biosecurity plan, water and feed management, confinement of the animals, rodent control, vaccination of cats and livestock) can contribute to the control of *T. gondii* in livestock by reducing the level of exposure to the infective stages from the environment [1,23] but sheep farming adds more complexity due to the outdoor pasture. Moreover, due to the incapability of traditional meat inspection tools of detecting microscopic cysts of *T. gondii* in the tissues of slaughtered animals and to the limitation of diagnostic methods, a risk-based approach (risk categorization of farms, followed by intervention, where appropriate) and additional efforts in the education of all stakeholders are considered more effective measures against *T. gondii* infection [23].

Considering this context, the study aims to provide further understanding on *T. gondii* transmission pathways and on the contribution of ovine meat as a potential source for human infection. In more detail, the study provides additional evidence regarding: (1) the occurrence of *T. gondii* in sheep muscles and (2) the strength of correlation between serological positivity and presence of the parasite in sheep. 

## 2. Materials and Methods

### 2.1. Experimental Design

The study design is described in detail in previous published research, where the same research group investigated the *T. gondii* seroprevalence of young and adult sheep from central Italy and potential environmental risk factors [4]. In brief, 405 sheep, from 91 farms, were randomly selected at the slaughterhouse level: a blood sample was collected from each animal and the serum was tested to detect the presence of immunoglobulin G (IgG) for *T. gondii* with an enzyme-linked immunosorbent assay (ELISA). The research found a seroprevalence of 43.7% and 62.8%, respectively, for young and adult sheep [4]. 

Furthermore, for each animal, the heart (whole organ) and part of diaphragm were also collected from the carcasses (in addition to the blood samples). Later, the same organs from 349 of them (158 young sheep and 191 adults, randomly selected) were tested with a real-time PCR to research the presence of *T. gondii*. A portion of the interventricular septum was taken from 130 hearts (28 young and 102 adult sheep) for a histological test to identify the presence of tissue cysts. 

### 2.2. Biomolecular Analysis

#### 2.2.1. DNA Extraction

DNA extraction from the interventricular septum and heart was performed using a QIAamp DNA Mini Kit (Qiagen, Hilden, Germany). The standard protocol was modified in order to obtain a complete lysis of the matrix. First, 25 mg of tissue was cut into small pieces and placed in a 2.0 mL microcentrifuge tube. ATL buffer (360 μL) and proteinase K (40 μL) were added, and the mixture was vortexed and incubated at 70 °C for 3 h. Buffer AL (200 μL) was added, vortexed for 15 s and incubated at 70 °C for 30 min. Ethanol (200 μL, 96–100%) was added and vortexed for 15 s. The mixture was then pipetted onto the QIAamp Mini spin column and centrifuged at 8000 rpm for 1 min. The flow-through and collection tube were discarded. The QIAamp Mini spin column was placed in a new 2 mL collection tube, and Buffer AW1 (500 μL) was added, followed by centrifugation at 8000 rpm for 1 min. The flow-through and collection tube were discarded. The QIAamp Mini spin column was placed in a new 2 mL collection tube, and Buffer AW2 (500 μL) was added. It was then centrifuged at 14,000 rpm for 3 min. The flow-through and collection tube were discarded. The QIAamp Mini spin column was placed in a new 1.5 mL microcentrifuge tube, and Buffer AE (200 μL) was added. The mixture was incubated at room temperature for 1 min. Finally, it was centrifuged at 8000 rpm for 1 min to elute the DNA. 

#### 2.2.2. DNA Amplification

Amplification reactions for *T. gondii* were performed with commercial kits (Bio-X Diagnostics S.A., Rochefort, Belgium) and carried out in a Rotor Gene Q (Qiagen, Hilden, Germany) with the following thermal profile: 2 min 50 °C; 10 min 95 °C followed by 45 cycles of 15 s at 95 °C and 1 min at 60 °C. Determining the presence or absence of pathogen DNA was carried out based on the amplification of the target sequence (not declared by the producer) and was visualized on the amplification plot generated by the Rotor Gene Q software (version 2.1.0; build 9). The PCR assay includes an internal control (IC) with high specificity that provides information on the presence of inhibitors in the analyzed samples and the overall success of the qPCR. 

The limit of detection (LOD) was not declared by the producer. 

### 2.3. Histological Test

Samples of interventricular septum (*n* = 130) were fixed in 10% buffered formalin, embedded in paraffin, cut into 5 µm thick pieces and then stained with hematoxylin and eosin for microscopic examination. 

### 2.4. Statistical Analysis

Results of the biomolecular and histological tests were recorded in an Excel spreadsheet (version 2016, Microsoft Corporation, Redmond, WA, USA) and described using frequencies and percentages (Clopper–Pearson confidential intervals) using R (version 4.1.1). The dataset was integrated with the result of the serological tests derived from the previous survey [4]. 

Cohen’s kappa coefficient was calculated to measure the agreement between the results of serological and biomolecular or histological tests. Records of PCR-tested animals found to be serologically dubious during the previous research were excluded from this calculation (*n* = 4).

## 3. Results

The presence of *T. gondii* DNA was detected in the heart and/or the diaphragm taken from 13 out of the 349 tested sheep (3.7%, 95% CI (2.0 to 6.3%)). All these PCR-positive subjects were adults (13/191, 6.8% 95% CI (3.7 to 11.4%)). In almost all of them (*n* = 12), *T. gondii* DNA was more frequently found in the heart than the diaphragm (*n* = 4) (Table 1). Few sheep (*n* = 3) had both a heart and diaphragm containing *T. gondii*. The presence of *T. gondii* only in the diaphragm happened only for one sheep.

Based on the serological findings previously reported [4], all PCR-positive sheep were seropositive apart from one animal. However, only 6.6% (12/181) of seropositive sheep were confirmed positive after the biomolecular test was applied to both heart and diaphragm. This percentage rose to 10.2% (12/117) when we considered only adult sheep (Table 2).

The agreement between the ELISA technique (serological status) and biomolecular technique (presence of *T. gondii* DNA in heart and/or diaphragm) was low (kappa = 0.06, 95% CI (0.02 to 0.09)) when the results referred to all sheep (young and adult animals) (Table 2). Cohen’s kappa coefficient decreased to 0.05 and 0.02 when we considered only one tissue at time (heart and diaphragm, respectively). If we took into account only the adult sheep, the Cohen’s kappa coefficient slightly increased to 0.07 (95% CI (0.02 to 0.12)).

The histological tests did not reveal the presence of *T. gondii* tissue cysts in any of the examined portions of interventricular septum (*n* = 130), although 68 of them were from seroreactive animals and 9 from PCR-positive ones.

## 4. Discussion

This study provided further information regarding the occurrence of *T. gondii* in the muscle of young and adult sheep and confirmed that the detection rate of the parasite with indirect techniques (serology) in sheep is higher than those determined with biomolecular techniques, at least following the described sampling design and experimental protocol. As presented by previous research, our results showed a low agreement between the two methods [24,25,26].

Several studies investigated the occurrence of *T. gondii* DNA in cardiac and/or skeleton muscle from sheep during recent decades. Dubey et al. [27] reported twenty surveys performed between 2009 and 2020 with a positivity rate that varied between 1.7% and 50% [26,27,28]. More recent studies found an occurrence rate included in this range (from 2% to 32%) [17,29,30,31,32].

The detection rate of *T. gondii* DNA from sheep muscle reported by our study (3.7%) is consistent with these intervals and close to the values reported by two previous studies conducted in Italy. Indeed, Pepe et al. [17] found 1 positive sample (heart) out of 50 (2%) from sheep at slaughterhouse while, after a preliminary screening with a serological method, Gazzonis et al. [33] detected *T. gondii* DNA in muscle samples (heart or diaphragm) from 15 out of 227 sheep (6.6%). Vismarra et al. [34] found *T. gondii* in a higher proportion of hearts (23.8%) but they experimented with a particular protocol that implied an “in vitro” cultivation of the parasite before performing a real-time PCR which could have improved the assay performance.

Numerous factors may have been responsible for the wide range of values related to *T. gondii* occurrence in sheep muscle. For example, it is known that the infection rate of *T. gondii* can significantly differ among geographic areas as well as certain farming practices/conditions, like adoption of an extensive rearing system or the presence of felids in the surroundings, which can increase the opportunities of infection of farmed animals [35]. Also, some environmental and climate conditions, like the presence of water bodies near the farming area or persistent mild temperatures, have been associated with a higher infection rate of intermediate hosts because they may be responsible for a wider spread of the oocysts and/or a longer period of their survival [4,36].

The differences in the study design can further explain the remarkable differences observed between the studies. For example, the age of the selected animals is considered an important factor as the longer their lifespan, the higher the probability that they will be infected [35,37]. In this context, we did not find *T. gondii* DNA in young animals while we found a high level of seropositivity [4] and other previous studies extracted genetic material of the parasite from lamb muscles [38,39]. However, the young animals (less than one year) we randomly selected during the sampling sessions at the slaughterhouse were generally very young as lamb meat consumed in Italy is habitually from non-weaned animals. Then, we can hypothesize that in several cases the presence of the antibodies was indeed due to passive immunity (which lasts around three months) and not to actual contact with the pathogen [26,29,40]. Anyway, even considering this aspect, young animals seemed less likely to harbor tissue cysts of *T. gondii* than adults.

The choice of the muscles sampled for the laboratory tests might also affect the probability of detection of *T. gondii* [20]. Some muscles could present a higher concentration of tissue cysts, so such uneven distribution of the parasite in the carcass makes it more complicated to compare the results from the different surveys [39,41,42]. Our results showed that *T. gondii* tissue cysts are more easy to find in cardiac tissue than diaphragm although this muscle was considered as a predilection site for the parasite among the skeletal muscles [20].

The type of molecular technique and the amount of muscle used for the DNA extraction are important factors that can affect the probability of detecting *T. gondii* DNA. Many kinds of molecular assays have been developed to detect the parasite in small ruminants, which vary in molecular target, DNA extraction protocol and amplification reaction conditions [43]. All these elements may significantly affect the performance of a diagnostic test and, consequently, the apparent prevalence in animals [20]. Even the amount of muscle that is examined, which varied between the studies, can play an important role in this sense. Considering that the distribution of the *T. gondii* tissue cysts is likely uneven in a muscle, the amount of tissue that is tested might substantially increase or decrease the likelihood of parasite detection. The relatively low amount that was analyzed in our study (25 mg) could contribute to explaining the lower recovery rate observed from the seropositive sheep compared to other studies.

We found a low concordance between serological and biomolecular tests, even when we considered only adult animals and combined results of both organs (heart and diaphragm together). Few studies have explored the relationship between seropositivity of a sheep and the presence of *T. gondii* DNA in its muscles so far, but a limited correlation is generally reported. Yousefvand et al. (2021) found a poor correlation between ELISA and PCR tests performed on samples from heart and meat (femur musculature) (kappa = 0.24 and 0.21, respectively) [30].

A similar level of agreement (MAT and PCR) was reported from Iranian researchers who tested blood and hearts from sheep sampled at a slaughterhouse (kappa = 0.33) [44]. Mazuz et al. (2023) obtained an even lower correlation testing 1.5 square centimeters from sheep’s heart (kappa calculated from the reported data is 0.08) [31]. However, Hamilton at al. (2015) found a moderate level of agreement between seropositivity (ELISA) and the presence of DNA (qPCR) in sheep heart (kappa calculated from the provided data = 0.67) [45]. This tendency to a low agreement could be mainly due to low values of recovery rate of *T. gondii* DNA in seropositive sheep. Indeed, all of the mentioned studies reported a recovery rate from muscle or heart that varied between 11% and 65% in the positive sheep. As explained in the previous paragraph, an uneven distribution of the tissue cysts in the carcass or in the muscle as well as a low amount of tested sample could impact on the likelihood of detecting genetic material of the parasite. On contrary, serological tests appear to be good predictors of negative muscle since, like in our study, the finding of the parasite DNA in heart or skeleton muscle from seronegative animals is an infrequent event [20,45,46].

Histological tests did not detect the presence of the parasite and this result seems to confirm the low sensitivity of this technique for muscle [20,46].

Several quantitative risk assessments were conducted to estimate the risk of human toxoplasmosis through the consumption of meat and some of them specifically regarded ovine meat products [9,21,22,47,48]. The results of this study are consistent with those of other research and confirm that assumption of a homogenous distribution of *T. gondii* in seropositive sheep carcasses might lead to inaccurate estimates and in particular to an overestimation of the risk [49]. This usual approach is even more conservative if we consider that biomolecular tests reveal the presence of *T. gondii* DNA but they do not show if the parasite is alive since bradyzoites in tissue cysts may not be vital after a certain period. However, the extent of this discrepancy between serological occurrence and the presence of the parasite in the different sheep muscles is still difficult to quantify with available data.

## 5. Conclusions

In conclusion, there is apparent disagreement between the results obtained through direct and indirect tests used to investigate *T. gondii* occurrence in sheep and the substantial variation among the studies, in terms of experimental design and laboratory protocols, does not allow clear identification of the responsible factors and their relevance. New studies are still needed to elucidate the actual occurrence of *T. gondii* in edible tissues in sheep populations and to better depict the distribution of the tissue cysts in seropositive animals. This new evidence would help to interpret the results of past and future serological surveys as well as the accuracy of predictive models that aim to estimate the risk of human toxoplasmosis linked to consumption of sheep meat. 

## Figures and Tables

**Table 1 animals-14-01432-t001:** Results of biomolecular tests applied to heart and diaphragm samples collected from 349 sheep (158 young sheep and 191 adults). The table reports the number of samples (*n*), the percentage by organ and age (%) and related confidence intervals.

Sample	PCR Result	Young Animals (*n*, %, 5% CI, 95% CI)	Adults (*n*, %, 5% CI, 95% CI)	All Animals (*n*, %, 5% CI, 95% CI)
Heart	Negative	158 (100%, 97.6%, 100%)	179 (93.7%, 89.2%, 96.7%)	337 (96.6%, 94.0%, 98%)
	Positive	0 (0%, 0%, 2.3%)	12 (6.3%, 3.2%, 10.7%)	12 (3.4%, 1.9%, 5.9%)
Diaphragm	Negative	158 (100%, 97.6%, 100%)	187 (97.9%, 94.7%, 99.4%)	345 (98.8%, 97%, 99.5%)
	Positive	0 (0%, 0%, 2.3%)	4 (2.1%, 0.5%, 5.2%)	4 (1.8%, 0.4%, 2.9%)
Total	Negative	316 (100%, 98.8%, 100%)	366 (95.8%, 93.2%, 97.5%)	682 (97.7%, 96.3%, 95.5%)
	Positive	0 (0%, 0%, 1.1%)	16 (4.2%, 2.4%, 6.7%)	16 (2.3%, 1.4%, 3.6%)

**Table 2 animals-14-01432-t002:** Number of sheep positive for serological tests, biomolecular tests (heart and/or the diaphragm) and related Cohen’s kappa coefficient.

Animal Category	*n*	ELISA	PCR		Total	Cohen’s Kappa
			Pos	Neg		
Young animals	156	Pos	0	64	64	0
		Neg	0	92	92	
Adult animals	189	Pos	12	105	117	0.07
		Neg	1	71	72	
All animals	345 *	Pos	12	169	181	0.06
		Neg	1	163	164	

* Animals dubious in serological tests were not included (*n* = 4 adult sheep which were negative in PCR tests).

## Data Availability

The raw data supporting the conclusions of this article will be made available by the authors on request.

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
