# Peer review of "Comparison of Direct and Indirect Detection of Toxoplasma gondii in Ovine Using Real-Time PCR, Serological and Histological Techniques"

_animals, 2024, doi:10.3390/ani14101432_

Round 1

Reviewer 1 Report

Comments and Suggestions for Authors

Authors used several techniques to estimate the proportion of sheep in Italy infected by T. gondii (ELISA, real-time PCR and histology). Methods used are valid; the outcome was expected and the conclusions are justified.

I have only minor comments:

Line 18 and entire manuscript:   I suggest to replace in the entire manuscript “cyst” by “tissue cyst”.

Line 47: I suggest to replace “eating” by “ingestion” or “feeding on”

Line 92-96: I think the statement  “On the other hand, risk assessments performed so far tend to be conservative and assume that muscle from all seropositive subjects are homogenously contaminated with a certain concentration bradyzoites. Nevertheless, this simplification might imply a relevant level of uncertainty associated with the risk of toxoplasmosis caused by meat consumption. “ is not true. There is a growing number of studies focussing this problem, including studies looking for predilection sites or studies which increase volume of tested samples to avoid to miss truly infected animals. I suggest to rephrase or to remove this statement.

Table 1: Please add 95% Confidence Intervals

Line 256: agreement is not correlation. Please rephrase

Comments on the Quality of English Language

NA

Author Response

Dear Reviewer,

the Authors thank you for the time you spent to revise our manuscript and the useful comments. We hope that all of them have been addressed. Please find the detailed responses in the attached word file.

Regards,

The Authors

Reviewer 2 Report

Comments and Suggestions for Authors

Dear Editor

The manuscript "Toxoplasma gondii in ovine: occurrence in muscle through real-time PCR in Italy and correlation with serological and histological techniques" is quite an interesting study. 

General Comments: The current title of the study is not clearly describing the theme of the study. It could be simplified for better understanding and to attract the global veterinarians.

For example, one suggestion is "Comparison of direct and indirect detection of Toxoplasma gondii in ovine using real-time PCRserological and histological techniques".

Specific Comments:

1. The introduction section should include global or regional annual sheep/ mutton production and consumption, this will strengthen the need of the project.

2. The Authors could have discussed the treatment regimes practiced in different regions along with it's impact on control of the T. gondii in sheep and goats.

3. The methods sections is well described, however that needs to have citations/ references.

4. The results sections lacks presentation of data in the form of tables/ figures. Only one table that does not describe sufficient results.

5. Authors can include the P-values and comparison of techniques to create good graphical image or table.

6. Discussion section is good.

Thanks and regards      

Author Response

(The authors gave the same response as above.)
